# Bayesian Control of Large MDPs with Unknown Dynamics in Data-Poor Environments

**Mahdi Imani**
Texas A&M University
College Station, TX, USA
m.imani88@tamu.edu

**Seyede Fatemeh Ghoreishi**
Texas A&M University
College Station, TX, USA
f.ghoreishi88@tamu.edu

**Ulisses M. Braga-Neto**
Texas A&M University
College Station, TX, USA
ulisses@ece.tamu.edu

## Abstract

We propose a Bayesian decision making framework for control of Markov Decision Processes (MDPs) with unknown dynamics and large, possibly continuous, state, action, and parameter spaces in data-poor environments. Most of the existing adaptive controllers for MDPs with unknown dynamics are based on the reinforcement learning framework and rely on large data sets acquired by sustained direct interaction with the system or via a simulator. This is not feasible in many applications, due to ethical, economic, and physical constraints. The proposed framework addresses the data poverty issue by decomposing the problem into an offline planning stage that does not rely on sustained direct interaction with the system or simulator and an online execution stage. In the offline process, parallel Gaussian process temporal difference (GPTD) learning techniques are employed for near-optimal Bayesian approximation of the expected discounted reward over a sample drawn from the prior distribution of unknown parameters. In the online stage, the action with the maximum expected return with respect to the posterior distribution of the parameters is selected. This is achieved by an approximation of the posterior distribution using a Markov Chain Monte Carlo (MCMC) algorithm, followed by constructing multiple Gaussian processes over the parameter space for efficient prediction of the means of the expected return at the MCMC sample. The effectiveness of the proposed framework is demonstrated using a simple dynamical system model with continuous state and action spaces, as well as a more complex model for a metastatic melanoma gene regulatory network observed through noisy synthetic gene expression data.

## 1 Introduction

Dynamic programming (DP) solves the optimal control problem for Markov Decision Processes (MDPs) with known dynamics and finite state and action spaces. However, in complex applications there is often uncertainty about the system dynamics. In addition, many practical problems have large or continuous state and action spaces. Reinforcement learning is a powerful technique widely used for adaptive control of MDPs with unknown dynamics [1]. Existing RL techniques developed for MDPs with unknown dynamics rely on data that is acquired via interaction with the system or via simulation. While this is feasible in areas such as robotics or speech recognition, in other applications such as medicine, materials science, and business, there is either a lack of reliable simulators or inaccessibility to the real system due to practical limitations, including cost, ethical, and physical considerations. For instance, recent advances in metagenomics and neuroscience call for the development of efficient intervention strategies for disease treatment. However, these systems are often modeled with MDPs with continuous state and action spaces, with limited access to expensive data. Thus, there is a need for control of systems with unknown dynamics and large or continuous state, action, and parameter spaces in data-poor environments.

**Related Work:** Approximate dynamic programming (ADP) techniques have been developed for problems in which the exact DP solution is not achievable. These include parametric and non-parametric reinforcement learning (RL) techniques for approximating the expected discounted reward over large or continuous state and action spaces. Parametric RL techniques include neural fitted Q-iteration [2], deep reinforcement learning [3], and kernel-based techniques [4]. A popular class of non-parametric RL techniques is Gaussian process temporal difference (GPTD) learning [5], which provides a Bayesian representation of the expected discounted return. However, all aforementioned methods involve approximate offline planning for MDPs with known dynamics or online learning by sustained direct interaction with the system or a simulator. The multiple model-based RL (MMRL) [6] is a framework that allows the extension of the aforementioned RL techniques to MDPs with unknown dynamics represented over a finite parameter space, and therefore cannot handle large or continuous parameter spaces.

In addition, there are several Bayesian reinforcement learning techniques in the literature [7]. For example, Bayes-adaptive RL methods assume a parametric family for the MDP transition matrix and simultaneously learn the parameters and policy. A closely related method in this class is the Beetle algorithm [8], which converts a finite-state MDP into a continuous partially-observed MDP (POMDP). Then, an approximate offline algorithm is developed to solve the POMDP. The Beetle algorithm is however capable of handling finite state and action spaces only. Online tree search approximations underlie a varied and popular class of Bayesian RL techniques [9–16]. In particular, the Bayes-adaptive Monte-Carlo planning (BAMCP) algorithm [16] has been shown empirically to outperform the other techniques in this category. This is due to the fact that BAMCP uses a rollout policy during the learning process, which effectively biases the search tree towards good solutions. However, this class of methods applies to finite-state MDPs with finite actions; application to continuous state and action spaces requires discretization of these spaces, rendering computation intractable in most cases of interest.

Lookahead policies are a well-studied class of techniques that can be used for control of MDPs with large or continuous state, action, and parameter spaces [17]. However, ignoring the long future horizon in their decision making process often results in poor performance. Other methods to deal with systems carrying other sources of uncertainty include [18, 19].

**Main Contributions:** The goal of this paper is to develop a framework for Bayesian decision making for MDPs with unknown dynamics and large or continuous state, action and parameter spaces in data-poor environments. The framework consists of offline and online stages. In the offline stage, samples are drawn from a prior distribution over the space of parameters. Then, parallel Gaussian process temporal difference (GPTD) learning algorithms are applied for Bayesian approximation of the expected discounted reward associated with these parameter samples. During the online process, a Markov Chain Monte Carlo (MCMC) algorithm is employed for sample-based approximation of the posterior distribution. For decision making with respect to the posterior distribution, Gaussian process regression over the parameter space based on the means and variances of the expected returns obtained in the offline process is used for prediction of the expected returns at the MCMC sample points. The proposed framework offers several benefits, summarized as follows:

- **Risk Consideration**: Most of the existing techniques try to estimate fixed values for approximating the expected Q-function and make a decision upon that, while the proposed method is capable of Bayesian representation of the Q-function. This allows risk consideration during action selection, which is required by many real-world applications, such as cancer drug design.

- **Fast Online Decision Making**: The proposed method is suitable for problems with tight time-limit constraints, in which the action should be selected relatively fast. Most of the computational effort spent by the proposed method is in the offline process. By contrast, the online process used by Monte-Carlo based techniques is often very slow, especially for large MDPs, in which a large number of trajectories must be simulated for accurate estimation of the Q-functions.

- **Continuous State/Action Spaces**: Existing Bayesian RL techniques can handle continuous state and action spaces to some extent (e.g., via discretization). However, the difficulty in picking a proper quantization rate, which directly impacts accuracy, and the computational intractability for large MDPs make the existing methods less attractive.

- **Generalization**: Another feature of the proposed method is the ability to serve as an initialization step for Monte-Carlo based techniques. In fact, if the expected error at each time point is large

(according to the Bayesian representation of the Q-functions), Monte-Carlo techniques can be employed for efficient online search using the available results of the proposed method.

- **Anytime Planning**: The Bayesian representation of the Q-function allows starting the online decision making process at anytime to improve the offline planning results. In fact, while the online planning is undertaken, the accuracy of the Q-functions at the current offline samples can be improved or the Q-functions at new offline samples from the posterior distribution can be computed.

## 2 Background

A Markov decision process (MDP) is formally defined by a 5-tuple $\langle \mathbb{S}, \mathbb{A}, T, R, \gamma \rangle$, where $\mathbb{S}$ is the *state space*, $\mathbb{A}$ is the *action space*, $T : \mathbb{S} \times \mathbb{A} \times \mathbb{S}$ is the *state transition probability function* such that $T(\mathbf{s}, \mathbf{a}, \mathbf{s}') = p(\mathbf{s}' \mid \mathbf{s}, \mathbf{a})$ represents the probability of moving to state $s'$ after taking action $\mathbf{a}$ in state $\mathbf{s}$, $R : \mathbb{S} \times \mathbb{A} \to \mathbb{R}$ is a bounded *reward function* such that $R(\mathbf{s}, \mathbf{a})$ encodes the reward earned when action $\mathbf{a}$ is taken in state $\mathbf{s}$, and $0 < \gamma < 1$ is a *discount factor*.

A deterministic stationary policy $\pi$ for an MDP is a mapping $\pi : \mathbb{S} \to \mathbb{A}$ from states to actions. The expected discounted reward function at state $\mathbf{s} \in \mathbb{S}$ after taking action $\mathbf{a} \in \mathbb{A}$ and following policy $\pi$ afterward is defined as:

$$Q^{\pi}(\mathbf{s}, \mathbf{a}) \,=\, E\left[ \sum_{t=0}^{\infty} \gamma^t R(\mathbf{s}_t, \mathbf{a}_t) \mid \mathbf{s}_0 = \mathbf{s}, \mathbf{a}_0 = \mathbf{a} \right]. \tag{1}$$

The optimal action-value function, denoted by $Q^*$, provides the maximum expected return $Q^*(\mathbf{s}, \mathbf{a})$ that can be obtained after executing action $\mathbf{a}$ in state $\mathbf{s}$. An optimal stationary policy $\pi^*$, which attains the maximum expected return for all states, is given by $\pi^*(\mathbf{s}) \,=\, \max_{\mathbf{a} \in \mathbb{A}} Q^*(\mathbf{s}, \mathbf{a})$.

An MDP is said to have known dynamics if the 5-tuple $\langle \mathbb{S}, \mathbb{A}, T, R, \gamma \rangle$ is fully specified, otherwise it is said to have unknown dynamics. For an MDP with known dynamics and finite state and action spaces, planning algorithms such as Value Iteration or Policy Iteration [20] can be used to compute the optimal policy offline. Several approximate dynamic programming (ADP) methods have been developed for approximating the optimal stationary policy over continuous state and action spaces. However, in this paper, we are concerned with large MDP with unknown dynamics in data-poor environments.

## 3 Proposed Bayesian Decision Framework

Let the unknown parts of the dynamics be encoded into a finite dimensional vector $\theta$, where $\theta$ takes value in a parameter space $\Theta \subset \mathbb{R}^m$. Notice that each $\theta \in \Theta$ specifies an MDP with known dynamics. Assuming $(\mathbf{a}_{0:k-1}, \mathbf{s}_{0:k})$ be the sequence of taken actions and observed states up to time step $k$ during the execution process, the proposed method selects an action according to:

$$\mathbf{a}_k \,=\, \underset{\mathbf{a} \in \mathbb{A}}{\operatorname{argmax}} \; \mathbb{E}_{\theta|\mathbf{s}_{0:k}, \mathbf{a}_{0:k-1}}[Q^*_{\theta}(\mathbf{s}_k, \mathbf{a})], \tag{2}$$

where the expectation is taken relative to the posterior distribution $p(\theta \mid \mathbf{s}_{0:k}, \mathbf{a}_{0:k-1})$, and $Q^*_{\theta}$ characterizes the optimal expected return for the MDP associated with $\theta$.

Two main issues complicate finding the exact solution in (2). First, computation of the posterior distribution might not have a closed-form solution, and one needs to use techniques such as Markov-Chain Monte-Carlo (MCMC) for sample-based approximation of the posterior. Secondly, the exact computation of $Q^*_{\theta}$ for any given $\theta$ is not possible, due to the large or possibly continuous state and action spaces. However, for any $\theta \in \Theta$, the expected return can be approximated with one of the many existing techniques such as neural fitted Q-iteration [2], deep reinforcement learning [3], and Gaussian process temporal difference (GPTD) learning [5]. On the other hand, all the afore-mentioned techniques can be extremely slow over an MCMC sample that is sufficiently large to achieve accurate results. In sum, computation of the expected returns associated with samples of the posterior distribution during the execution process is not practical.

In the following paragraphs, we propose efficient offline and online planning processes capable of computing an approximate solution to the optimization problem in (2).

## 3.1 Offline Planner

The offline process starts by drawing a sample $\Theta^{\text{prior}} = \{\theta_i^{\text{prior}}\}_{i=1}^{N^{\text{prior}}} \sim p(\theta)$ of size $N^{\text{prior}}$ from the parameter prior distribution. For each sample point $\theta \in \Theta^{\text{prior}}$, one needs to approximate the optimal expected return $Q_\theta^*$ over the entire state and action spaces. We propose to do this by using Gaussian process temporal difference (GPTD) learning [5]. The detailed reasoning behind this choice will be provided when the online planner is discussed.

GP-SARSA is a GPTD algorithm that provides a Bayesian approximation of $Q_\theta^*$ for given $\theta \in \Theta^{\text{prior}}$. We describe the GP-SARSA algorithm over the next several paragraphs. Given a policy $\pi_\theta : \mathbb{S} \to \mathbb{A}$ for an MDP corresponding to $\theta$, the discounted return at time step $t$ can be written as:

$$U_\theta^{t,\pi_\theta}(\mathbf{s}_t, \mathbf{a}_t) = \mathbb{E}\left[ \sum_{r=t}^{\infty} \gamma^{r-t} R_\theta(\mathbf{s}_{r+1}, \mathbf{a}_{r+1}) \right], \tag{3}$$

where $\mathbf{s}_{r+1} \sim p\left(\mathbf{s}' \mid \mathbf{s}_r, \mathbf{a}_r = \pi_\theta(\mathbf{s}_r), \theta\right)$, and $U_\theta^{t,\pi_\theta}(\mathbf{s}_t, \mathbf{a}_t)$ is the expected accumulated reward for the system corresponding to parameter $\theta$ obtained over time if the current state and action are $\mathbf{s}_t$ and $\mathbf{a}_t$ and policy $\pi_\theta$ is followed afterward.

In the GPTD method, the expected discounted return $U_\theta^{t,\pi_\theta}(\mathbf{s}_t, \mathbf{a}_t)$ is approximated as:

$$U_\theta^{t,\pi_\theta}(\mathbf{s}_t = \mathbf{s}, \mathbf{a}_t = \mathbf{a}) \approx Q_\theta^{\pi_\theta}(\mathbf{s}, \mathbf{a}) + \Delta Q_\theta^{\pi_\theta}, \tag{4}$$

where $Q_\theta^{\pi_\theta}(\mathbf{s}, \mathbf{a})$ is a *Gaussian process* [21] over the space $\mathbb{S} \times \mathbb{A}$ and $\Delta Q_\theta^{\pi_\theta}$ is a zero-mean Gaussian residual with variance $\sigma_q^2$. A zero-mean Gaussian process is usually considered as a prior:

$$Q_\theta^{\pi_\theta}(\mathbf{s}, \mathbf{a}) = \mathcal{GP}\left(\mathbf{0}, k_\theta\left((\mathbf{s}, \mathbf{a}), (\mathbf{s}', \mathbf{a}')\right)\right), \tag{5}$$

where $k_\theta(\cdot, \cdot)$ is a real-valued kernel function, which encodes our prior belief on the correlation between $(\mathbf{s}, \mathbf{a})$ and $(\mathbf{s}', \mathbf{a}')$. One possible choice is considering decomposable kernels over the state and action spaces: $k_\theta\left((\mathbf{s}, \mathbf{a}), (\mathbf{s}', \mathbf{a}')\right) = k_{\mathbb{S},\theta}\left(\mathbf{s}, \mathbf{s}'\right) \times k_{\mathbb{U},\theta}\left(\mathbf{a}, \mathbf{a}'\right)$. A proper choice of the kernel function depends on the nature of the state and action spaces, e.g., whether they are finite or continuous.

Let $\mathbf{B}_t^\theta = [(\mathbf{s}_0, \mathbf{a}_0), \dots, (\mathbf{s}_t, \mathbf{a}_t)]^T$ be the sequence of observed joint state and action pairs simulated by a policy $\pi_\theta$ from an MDP corresponding to $\theta$, with the corresponding immediate rewards $\mathbf{r}_t^\theta = [R_\theta(\mathbf{s}_0, \mathbf{a}_0), \dots, R_\theta(\mathbf{s}_t, \mathbf{a}_t)]^T$. The posterior distribution of $Q_\theta^{\pi_\theta}(\mathbf{s}, \mathbf{a})$ can be written as [5]:

$$Q_\theta^{\pi_\theta}(\mathbf{s}, \mathbf{a}) \mid \mathbf{r}_t^\theta, \mathbf{B}_t^\theta \sim \mathcal{N}\left(\overline{Q}_\theta(\mathbf{s}, \mathbf{a}), \text{cov}_\theta\left((\mathbf{s}, \mathbf{a}), (\mathbf{s}, \mathbf{a})\right)\right), \tag{6}$$

where

$$\overline{Q}_\theta(\mathbf{s}, \mathbf{a}) = \mathbf{K}_{(\mathbf{s},\mathbf{a}),\mathbf{B}_t^\theta} \mathbf{H}_t^T (\mathbf{H}_t \mathbf{K}_{\mathbf{B}_t^\theta,\mathbf{B}_t^\theta} \mathbf{H}_t^T + \sigma_q^2 \mathbf{H}_t \mathbf{H}_t^T)^{-1} \mathbf{r}_t^\theta,$$

$$\text{cov}_\theta((\mathbf{s},\mathbf{a}), (\mathbf{s},\mathbf{a})) = k_\theta((\mathbf{s}, \mathbf{a}), (\mathbf{s}, \mathbf{a})) - \mathbf{K}_{(\mathbf{s},\mathbf{a}),\mathbf{B}_t^\theta} \mathbf{H}_t^T (\mathbf{H}_t \mathbf{K}_{\mathbf{B}_t^\theta,\mathbf{B}_t^\theta} \mathbf{H}_t^T + (\sigma_\theta^q)^2 \mathbf{H}_t \mathbf{H}_t^T)^{-1} \mathbf{H}_t \mathbf{K}_{(\mathbf{s},\mathbf{a}),\mathbf{B}_t^\theta}^T$$

$$\tag{7}$$

with

$$\mathbf{H}_t = \begin{bmatrix} 1 & -\gamma & 0 & \dots & 0 & 0 \\ 0 & 1 & -\gamma & \dots & 0 & 0 \\ \vdots & \vdots & \vdots & \ddots & \vdots & \vdots \\ 0 & 0 & 0 & \dots & 1 & -\gamma \\ 0 & 0 & 0 & \dots & 0 & 1 \end{bmatrix}, \mathbf{K}_{\mathbf{B},\mathbf{B}'} = \begin{bmatrix} k_\theta((\mathbf{s}_0, \mathbf{a}_0), (\mathbf{s}_0', \mathbf{a}_0')) & \dots & k_\theta((\mathbf{s}_0, \mathbf{a}_0), (\mathbf{s}_n', \mathbf{a}_n')) \\ \vdots & \ddots & \vdots \\ k_\theta((\mathbf{s}_m, \mathbf{a}_m), (\mathbf{s}_0', \mathbf{a}_0')) & \dots & k_\theta((\mathbf{s}_m, \mathbf{a}_m), (\mathbf{s}_n', \mathbf{a}_n')) \end{bmatrix},$$

$$\tag{8}$$

for $\mathbf{B} = [(\mathbf{s}_0, \mathbf{a}_0), \dots, (\mathbf{s}_m, \mathbf{a}_m)]^T$ and $\mathbf{B}' = [(\mathbf{s}_0', \mathbf{a}_0'), \dots, (\mathbf{s}_n', \mathbf{a}_n')]^T$.

The hyper-parameters of the kernel function can be estimated by maximizing the likelihood of the observed reward [22]:

$$\mathbf{r}_t^\theta \mid \mathbf{B}_t^\theta \sim \mathcal{N}\left(\mathbf{0}, \mathbf{H}_t\left(\mathbf{K}_{\mathbf{B}_t^\theta,\mathbf{B}_t^\theta} + (\sigma_q^\theta)^2 \mathbf{I}_t \mathbf{H}_t^T\right)\right), \tag{9}$$

where $\mathbf{I}_t$ is the identity matrix of size $t \times t$.

The choice of policy for gathering data has significant impact on the proximity of the estimated discounted return to the optimal one. A well-known option, which uses Bayesian representation of the expected return and adaptively balances exploration and exploitation, is given by [22]:

$$\pi_\theta(\mathbf{s}) = \underset{\mathbf{a} \in \mathbb{A}}{\arg\max}\ q_a,\ \ q_a \sim \mathcal{N}\left(\overline{Q}_\theta(\mathbf{s}, \mathbf{a}), \text{cov}_\theta\left((\mathbf{s}, \mathbf{a}), (\mathbf{s}, \mathbf{a})\right)\right). \tag{10}$$

The GP-SARSA algorithm approximates the expected return by simulating several trajectories based on the above policy. Running $N^{\text{prior}}$ parallel GP-SARSA algorithms for each $\theta \in \Theta^{\text{prior}}$ leads to $N^{\text{prior}}$ near-optimal approximations of the expected reward functions.

## 3.2 Online Planner

Let $\hat{Q}_\theta(\mathbf{s}, \mathbf{a})$ be the Gaussian process approximating the optimal Q-function for any $\mathbf{s} \in \mathbb{S}$ and $\mathbf{a} \in \mathbb{A}$ computed by a GP-SARSA algorithm associated with parameter $\theta \in \Theta^{\text{prior}}$. One can approximate (2) as:

$$\mathbf{a}_k \approx \underset{\mathbf{a} \in \mathbb{A}}{\operatorname{argmax}} \, \mathbb{E}_{\theta | \mathbf{s}_{0:k}, \mathbf{a}_{0:k-1}} \left[ \mathbb{E} \left[ \hat{Q}_\theta(\mathbf{s}_k, \mathbf{a}) \right] \right] = \underset{\mathbf{a} \in \mathbb{A}}{\operatorname{argmax}} \, \mathbb{E}_{\theta | \mathbf{s}_{0:k}, \mathbf{a}_{0:k-1}} \left[ \overline{Q}_\theta(\mathbf{s}_k, \mathbf{a}) \right] . \tag{11}$$

While the value of $\overline{Q}_\theta(\mathbf{s}_k, \mathbf{a})$ at values $\theta \in \Theta^{\text{prior}}$ drawn from the prior distribution is available, the expectation in (11) is over the posterior distribution.

Rather than restricting ourselves to parametric families, we compute the expectation in (11) by a Markov Chain Monte-Carlo (MCMC) algorithm for generating i.i.d. sample values from the posterior distribution. For simplicity, and without loss of generality, we employ the basic Metropolis Hastings MCMC [23] algorithm. Let the last accepted MCMC sample in the sequence of samples be $\theta_j^{\text{post}}$, generated at the $j$-th iteration. A candidate MCMC sample point $\theta^{\text{cand}}$ is drawn according to a symmetric proposal distribution $q(\theta \mid \theta_j^{\text{post}})$. The candidate MCMC sample point $\theta^{\text{cand}}$ is accepted with probability $\alpha$ given by:

$$\alpha = \min \left\{ 1, \frac{p(\mathbf{s}_{0:k}, \mathbf{a}_{0:k-1} \mid \theta^{\text{cand}}) \, p(\theta^{\text{cand}})}{p(\mathbf{s}_{0:k}, \mathbf{a}_{0:k-1} \mid \theta_j^{\text{post}}) \, p(\theta_j^{\text{post}})} \right\} , \tag{12}$$

otherwise it is rejected, where $p(\theta)$ denotes the prior probability of $\theta$. Accordingly, the $(j+1)$th MCMC sample point is:

$$\theta_{j+1}^{\text{post}} = \begin{cases} \theta^{\text{n}} & \text{with probability } \alpha \\ \theta_j^{\text{post}} & \text{otherwise} \end{cases} \tag{13}$$

Repeating this process leads to a sequence of MCMC sample points. The positivity of the proposal distribution (i.e. $q(\theta \mid \theta_j^{\text{post}}) > 0$, for any $\theta_j^{\text{post}}$) is a sufficient condition for ensuring an ergodic Markov chain whose steady-state distribution is the posterior distribution $p(\theta \mid \mathbf{s}_{0:k}, a_{0:k-1})$ [24]. Removing a fixed number of initial "burn-in" sample points, the MCMC sample $\Theta^{\text{post}} = (\theta_1^{\text{post}}, \ldots, \theta_{N^{\text{post}}}^{\text{post}})$ is approximately a sample from the posterior distribution.

The last step towards the computation of (11) is the approximation of the mean of the predicted expected return $\overline{Q}_\theta(.,.)$ at values of the MCMC sample $\Theta^{\text{post}}$. We take advantage of the Bayesian representation of the expected return computed by the offline GP-SARSAs for this, as described next.

Let $\mathbf{f}_{\mathbf{s}_k}^{\mathbf{a}} = [\overline{Q}_{\theta_1^{\text{prior}}}(\mathbf{s}_k, \mathbf{a}), \ldots, \overline{Q}_{\theta_{N^{\text{prior}}}^{\text{prior}}}(\mathbf{s}_k, \mathbf{a})]^T$ and $\mathbf{v}_{\mathbf{s}_k}^{\mathbf{a}} = [\operatorname{cov}_{\theta_1^{\text{prior}}}((\mathbf{s}_k, \mathbf{a}), (\mathbf{s}_k, \mathbf{a})), \ldots, \operatorname{cov}_{\theta_{N^{\text{prior}}}^{\text{prior}}}((\mathbf{s}_k, \mathbf{a}), (\mathbf{s}_k, \mathbf{a}))]^T$ be the means and variances of the predicted expected returns computed based on the results of offline GP-SARSAs at current state $\mathbf{s}_k$ for a given action $\mathbf{a} \in \mathbb{A}$. This information can be used for constructing a Gaussian process for predicting the expected return over the MCMC sample:

$$\begin{bmatrix} \overline{Q}_{\theta_1^{\text{post}}}(\mathbf{s}_k, \mathbf{a}) \\ \vdots \\ \overline{Q}_{\theta_{N^{\text{post}}}^{\text{post}}}(\mathbf{s}_k, \mathbf{a}) \end{bmatrix} = \mathbf{\Sigma}_{\Theta^{\text{post}}, \Theta^{\text{prior}}} \left( \mathbf{\Sigma}_{\Theta^{\text{prior}}, \Theta^{\text{prior}}} + \operatorname{Diag}(\mathbf{v}_k^{\mathbf{a}}) \right)^{-1} \mathbf{f}_{\mathbf{s}_k}^{\mathbf{a}}, \tag{14}$$

where

$$\mathbf{\Sigma}_{\Theta_m, \Theta_n} = \begin{bmatrix} k(\theta_1, \theta_1') & \ldots & k(\theta_1, \theta_n') \\ \vdots & \ddots & \vdots \\ k(\theta_m, \theta_n') & \ldots & k(\theta_m, \theta_n') \end{bmatrix},$$

for $\Theta_m = \{\theta_1, \ldots, \theta_m\}$ and $\Theta_n = \{\theta'_1, \ldots, \theta'_n\}$, and $k(\theta, \theta')$ denotes the correlation between sample points in the parameter space. The parameters of the kernel function can be inferred by maximizing the marginal likelihood:

$$\mathbf{f}_k^{\mathbf{a}} \mid \Theta^{\text{prior}} \sim \mathcal{N}\left(\mathbf{0}, \boldsymbol{\Sigma}_{\Theta^{\text{prior}}, \Theta^{\text{prior}}} + \text{Diag}(\mathbf{v}_k^{\mathbf{a}})\right) . \tag{15}$$

The process is summarized in Figure 1(a). The red vertical lines represent the expected returns at sample points from the posterior. It can be seen that only a single offline sample point is in the area covered by the MCMC samples, which illustrates the advantage of the constructed Gaussian process for predicting the expected return over the posterior distribution.

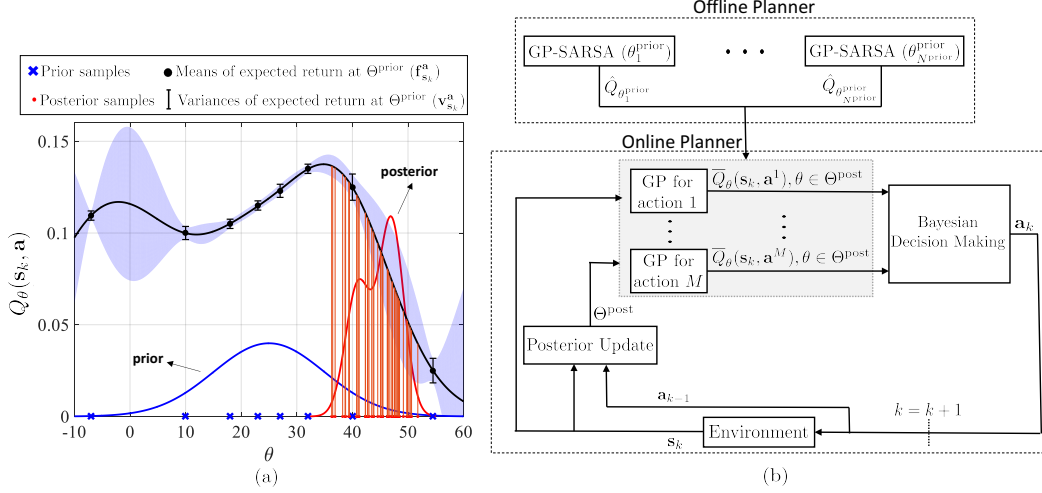

Figure 1: (a) Gaussian process for prediction of the expected returns at posterior sample points based on prior sample points. (b) Proposed framework.

The GP is constructed for any given $\mathbf{a} \in \mathbb{A}$. For a large or continuous action space, one needs to draw a finite set of actions $\{a^1, \ldots, a^M\}$ from the space, and compute $\overline{Q}_\theta(\mathbf{s}_k, \mathbf{a})$ for $\mathbf{a} \in \{a^1, \ldots, a^M\}$ and $\theta \in \Theta^{\text{post}}$. It should be noted that the uncertainty in the expected return of the offline sample points is efficiently taken into account for predicting the mean expected error at the MCMC sample points. Thus, equation (11) can be written as:

$$\mathbf{a}_k \approx \underset{\mathbf{a} \in \mathbb{A}}{\text{argmax}} \, \mathbb{E}_{\theta \mid \mathbf{s}_{0:k}, \mathbf{a}_{0:k-1}} \left[\overline{Q}_\theta(\mathbf{s}_k, \mathbf{a})\right] \approx \underset{\mathbf{a} \in \{a^1, \ldots, a^M\}}{\text{argmax}} \frac{1}{N^{\text{post}}} \sum_{\theta \in \Theta^{\text{post}}} \overline{Q}_\theta(\mathbf{s}_k, \mathbf{a}) . \tag{16}$$

It is shown empirically in numerical experiments that as more data are observed during execution, the proposed method becomes more accurate, eventually achieving the performance of a GP-SARSA trained on data from the true system model. The entire proposed methodology is summarized in Algorithm 1 and Figure 1(b) respectively.

Notice that the values of $N^{\text{prior}}$ and $N^{\text{post}}$ should be chosen based on the size of the MDP, the availability of computational resources, and presence of time constraints. Indeed, large $N^{\text{prior}}$ means that larger parameter samples must be obtained in the offline process, while large $N^{\text{post}}$ is associated with larger MCMC samples in the posterior update step.

## 4 Numerical Experiments

The numerical experiments compare the performance of the proposed framework with two other methods: 1) Multiple Model-based RL (MMRL) [6]: the parameter space in this method is quantized into a finite set $\Theta^{\text{quant}}$ according to its prior distribution and the results of offline parallel GP-SARSA algorithms associated with this set are used for decision making during the execution process via: $\mathbf{a}_k^{\text{MMRL}} = \text{argmax}_{\mathbf{a} \in \mathbb{A}} \sum_{\theta \in \Theta^{\text{quant}}} \overline{Q}_\theta(\mathbf{s}_k, \mathbf{a}_k = \mathbf{a}) P(\theta \mid \mathbf{s}_{0:k}, \mathbf{a}_{0:k-1})$. 2) One-step lookahead policy [17]: this method selects the action with the highest expected immediate reward: $\mathbf{a}_k^{\text{seq}} = \text{argmax}_{\mathbf{a} \in \mathbb{A}} \mathbb{E}_{\theta \mid \mathbf{s}_{0:k}, \mathbf{a}_{0:k-1}} [R(\mathbf{s}_k, \mathbf{a}_k = \mathbf{a})]$. As a baseline for performance, the results of the GP-SARSA algorithm tuned to the true model are also displayed.

---

**Algorithm 1** Bayesian Control of Large MDPs with Unknown Dynamics in Data-Poor Environments.

---

*Offline Planning*

1: Draw $N^{\text{prior}}$ parameters from the prior distribution: $\Theta^{\text{prior}} = \{\theta_1, \ldots, \theta_{N^{\text{prior}}}\} \sim p(\theta)$.

2: Run $N^{\text{prior}}$ parallel GP-SARSAs:

$$\hat{Q}_\theta \leftarrow \text{GP-SARSA}(\theta), \, \theta \in \Theta^{\text{prior}}.$$

*Online Planning*

3: Initial action selection:

$$\mathbf{a}_0 = \arg\max_{\mathbf{a} \in \mathbb{A}} \frac{1}{N^{\text{prior}}} \sum_{\theta \in \Theta^{\text{prior}}} \overline{Q}_\theta(\mathbf{s}_0, \mathbf{a}).$$

4: **for** $k = 1, \ldots$ **do**

5:     Take action $\mathbf{a}_{k-1}$, record the new state $\mathbf{s}_k$.

6:     Given $\mathbf{s}_{0:k}, \mathbf{a}_{0:k-1}$, run MCMC and collect $\Theta_k^{\text{post}}$.

7:     **for** $\mathbf{a} \in \{\mathbf{a}^1, \ldots, \mathbf{a}^M\}$ **do**

8:         Record the means and variances of offline GPs at $(\mathbf{s}_k, \mathbf{a})$:

$$\mathbf{f}_{\mathbf{s}_k}^{\mathbf{a}} = [\overline{Q}_{\theta_1}(\mathbf{s}_k, \mathbf{a}), \ldots, \overline{Q}_{\theta_{N^{\text{prior}}}^{\text{prior}}}(\mathbf{s}_k, \mathbf{a})]^T,$$

$$\mathbf{v}_{\mathbf{s}_k}^{\mathbf{a}} = [\text{cov}_{\theta_1}((\mathbf{s}_k, \mathbf{a}), (\mathbf{s}_k, \mathbf{a})), \ldots, \text{cov}_{\theta_{N^{\text{prior}}}^{\text{prior}}}((\mathbf{s}_k, \mathbf{a}), (\mathbf{s}_k, \mathbf{a}))]^T.$$

9:         Construct a GP using $\mathbf{f}_{\mathbf{s}_k}^{\mathbf{a}}, \mathbf{v}_{\mathbf{s}_k}^{\mathbf{a}}$ over $\Theta^{\text{prior}}$.

10:       Use the constructed GP to compute $\overline{Q}_\theta(\mathbf{s}_k, \mathbf{a})$, for $\theta \in \Theta^{\text{post}}$.

11:     **end for**

12:     Action selection:

$$\mathbf{a}_k = \arg\max_{\mathbf{a} \in \{\mathbf{a}^1, \ldots, \mathbf{a}^M\}} \frac{1}{N^{\text{post}}} \sum_{\theta \in \Theta^{\text{post}}} \overline{Q}_\theta(\mathbf{s}_k, \mathbf{a}).$$

13: **end for**

---

**Simple Continuous State and Action Example:** The following simple MDP with unknown dynamics is considered in this section:

$$s_k = \text{bound}[s_{k-1} - \theta s_{k-1}(0.5 - s_{k-1}) + 0.2a_{k-1} + n_k], \tag{17}$$

where $s_k \in \mathbb{S} = [0, 1]$ and $\mathbf{a}_k \in \mathbb{A} = [-1, 1]$ for any $k \geq 0$, $n_k \sim \mathcal{N}(0, 0.05)$, $\theta$ is the unknown parameter with true value $\theta^* = 0.2$ and bound maps the argument to the closest point in state space. The reward function is $R(s, a) = -10\,\delta_{s<0.1} - 10\,\delta_{s>0.9} - 2\,|a|$, so that the cost is minimum when the system is in the interval $[0.1, 0.9]$. The prior distribution is $p(\theta) \sim \mathcal{N}(0, 0.2)$. The decomposable squared exponential kernel function is used over the state and action spaces. The offline and MCMC sample sizes are 10 and 1000, respectively.

Figures 2(a) and (b) plot the optimal actions in the state and parameter spaces and the Q-function over state and action spaces for the true model $\theta^*$, obtained by GP-SARSA algorithms. It can be seen that the decision is significantly impacted by the parameter, especially in regions of the state space between 0.5 to 1. The Bayesian approximation of the Q-function is represented by two surfaces that define 95%-confidence intervals for the expected return. The average reward per step over 100 independent runs starting from different initial states are plotted in Figure 2(c). As expected, the maximum average reward is obtained by the GP-SARSA associated with the true model. The proposed framework significantly outperforms both MMRL and one-step lookahead techniques. One can see that the average reward for the proposed algorithm converges to the true model results after less than 20 actions while the other methods do not. The very poor performance of the one-step lookahead method is due to the greedy heuristics involved in its decision making process, which do not factor in long-term rewards.

**Melanoma Gene Regulatory Network**: A key goal in genomics is to find proper intervention strategies for disease treatment and prevention. Melanoma is the most dangerous form of skin cancer, the gene-expression behavior of which can be represented through the Boolean activities of 7 genes displayed in Figure 3. Each gene expression can be 0 or 1, corresponding to gene inactivation or activation, respectively. The gene states are assumed to be updated at each discrete time through

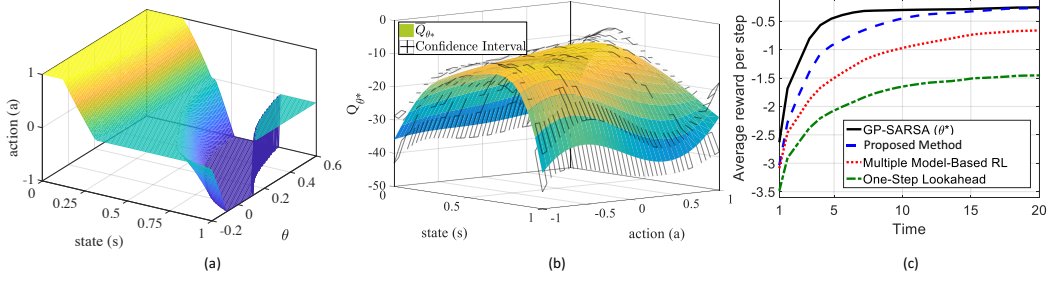

$$(a) \qquad (b) \qquad (c)$$

Figure 2: Small example results.

the following nonlinear signal model:

$$\mathbf{x}_k \;=\; \mathbf{f}\left(\mathbf{x}_{k-1}\right) \,\oplus\, \mathbf{a}_{k-1} \,\oplus\, \mathbf{n}_k \,, \tag{18}$$

where $\mathbf{x}_k = \left[\mathrm{WNT5A}_k, \mathrm{pirin}_k, \mathrm{S100P}_k, \mathrm{RET1}_k, \mathrm{MART1}_k, \mathrm{HADHB}_k, \mathrm{STC2}_k\right]$ is the state vector at time step $k$, action $\mathbf{a}_{k-1} \in \mathbb{A} \subset \{0,1\}^7$, such that $\mathbf{a}_{k-1}(i) = 1$ flips the state of the $i$th gene, $\mathbf{f}$ is the Boolean function displayed in Table 1, in which the $i$th binary string specifies the output value for the given input genes, "$\oplus$" indicates component-wise modulo-2 addition and $\mathbf{n}_k \in \{0,1\}^7$ is Boolean transition noise, such that $P(\mathbf{n}_k(i) = 1) = p$, for $i = 1, \ldots, 7$.

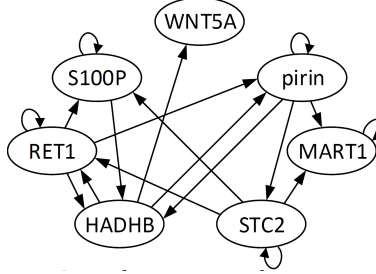

Figure 3: Melanoma regulatory network

Table 1: Boolean functions for the melanoma GRN.

| Genes | Input Gene(s) | Output |
|---|---|---|
| WNT5A | HADHB | 10 |
| pirin | prin, RET1,HADHB | 00010111 |
| S100P | S100P,RET1,STC2 | 10101010 |
| RET1 | RET1,HADHB,STC2 | 00001111 |
| MART1 | pirin,MART1,STC2 | 10101111 |
| HADHB | pirin,S100P,RET1 | 01110111 |
| STC2 | pirin,STC2 | 1101 |

In practice, the gene states are observed through gene expression technologies such as cDNA microarray or image-based assay. A Gaussian observation model is appropriate for modeling the gene expression data produced by these technologies:

$$\mathbf{y}_k(i) \;\sim\; \mathcal{N}\left(20\,\mathbf{x}_k(i) + \theta, 10\right) \,, \tag{19}$$

for $i = 1, \ldots, 7$; where parameter $\theta$ is the baseline expression in the inactivated state with the true value $\theta^* = 30$. Such a model is known as a partially-observed Boolean dynamical system in the literature [25, 26].

It can be shown that for any given $\theta \in \mathbb{R}$, the partially-observed MDP in (18) and (19) can be transformed into an MDP in a continuous belief space [27, 28]:

$$\begin{aligned}
\mathbf{s}_k \;&=\; \mathbf{g}(\mathbf{s}_{k-1}, \mathbf{a}_{k-1}, \theta) \\
&\propto\; p(\mathbf{y}_k \mid \mathbf{x}_k, \theta)\, P(\mathbf{x}_k \mid \mathbf{x}_{k-1}, \mathbf{a}_k)\, \mathbf{s}_{k-1} \,,
\end{aligned} \tag{20}$$

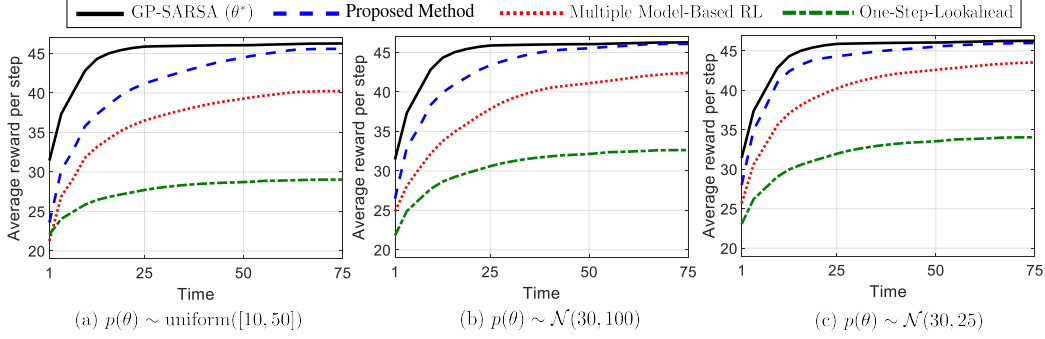

Figure 4: Melanoma gene regulatory network results.

where "$\propto$" indicates that the right-hand side must be normalized to add up to $1$. The belief state is a vector of length $128$ in a simplex of size $127$.

In [29, 30], the expression of WNT5A was found to be highly discriminatory between cells with properties typically associated with high metastatic competence versus those with low metastatic competence. Hence, an intervention that blocked the WNT5A protein from activating its receptor could substantially reduce the ability of WNT5A to induce a metastatic phenotype. Thus, we consider the following immediate reward function in belief space: $R(\mathbf{s}, \mathbf{a}) = 50 \sum_{i=1}^{128} \mathbf{s}(i)\, \delta_{\mathbf{x}^i(1)=0} - 10||\mathbf{a}||_1$. Three actions are available for controlling the system: $\mathbb{A} = \{[0,0,0,0,0,0,0], [0,0,0,1,0,0,0], [0,0,0,0,0,1,0]\}$.

The decomposable squared exponential and delta Kronecker kernel functions are used for Gaussian process regression over the belief state and action spaces, respectively. The offline and MCMC sample sizes are 10 and 3000, respectively. The average reward per step over 100 independent runs for all methods is displayed in Figure 4. Uniform and Gaussian distributions with different variances are used as prior distributions in order to investigate the effect of prior peakedness. As expected, the highest average reward is obtained for GP-SARSA tuned to the true parameter $\theta^*$. The proposed method has higher average reward than the MMRL and one-step lookahead algorithms. In fact, the expected return produced by the proposed method converges to the GP-SARSA tuned to the true parameter faster for peaked prior distributions. As more actions are taken, the performance of MMRL approaches, but not quite reaches, the baseline performance of the GP-SARSA tuned to the true parameter. The one-step lookahead method performs poorly in all cases as it does not account for long-term rewards in the decision making process.

## 5   Conclusion

In this paper, we introduced a Bayesian decision making framework for control of MDPs with unknown dynamics and large or continuous state, actions and parameter spaces in data-poor environments. The proposed framework does not require sustained direct interaction with the system or a simulator, but instead it plans offline over a finite sample of parameters from a prior distribution over the parameter space and transfers this knowledge efficiently to sample parameters from the posterior during the execution process. The methodology offers several benefits, including the possibility of handling large and possibly continuous state, action, and parameter spaces; data-poor environments; anytime planning; and dealing with risk in the decision making process.

## Acknowledgment

The authors acknowledge the support of the National Science Foundation, through NSF award CCF-1718924.

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
