[Reviews · NeurIPS 2018]

Reviewer 1



POST-REBUTTAL: Thank you for the clarifications. I've increased the overall score because the rebuttal made me think of this work as an interesting tradeoff between generality and the time when when computational cost investment has to be made (online vs. offline). At the same time, BAMCP has the significant advantage of handling more stochastic (as opposed to nearly deterministic, as in this paper) scenarios, and I would encourage you to work on extending your approach to them. ============================================================== The paper proposes a Bayesian RL method for MDPs with continuous states, action, and parameter spaces. Its key component is efficiently approximating the Q-value function expectations over the posterior belief over the parameter space, which is achieved by a combination of GPTD during the offline stage of the algorithm and MCMC during the online stage. The approach is evaluated against MMRL and greedy myopic action selection on two problems, one of which is a synthetic MDP and another is a problem of computing gene expression interventions for a regulatory network. To me, the paper's main novelty is the technique for approximating Q-value function expectations over parameter posterior given approximate Q-values for a sample of parameter values drawn from the prior. The approach has quite a few moving parts, and it's impressive to see them work in concert. At the same time, the paper would benefit greatly from a better explanation of its motivation/applicability and, on a related note, from crisper positioning w.r.t. related work. The lack of clarity in these aspects makes it difficult to determine potential existing alternatives (if any) to the proposed algorithm and its parts. More specifically: - Motivation. The submission's promise of handling decision-making problems with uncertain dynamics and continuous parameter spaces in data-poor environments isn't unique to the approach presented here. It is the motivation for Bayesian RL in general, and different BRL methods deliver on it varying degrees of success. The question is: what exactly distinguishes the proposed approach's capabilities from prior art in BRL? Is it the ability to handle continuous state spaces? This is a bit hard to believe, because given a sample of model parameters, i.e., a fully specified MDP, there is a range of methods that can solve such MDPs; discretizing the state space is one option. Same goes for continous action spaces. In fact, I don't see why this paper's method would have particular advantages in this regard, because despite its ability to approximate Q-value expectations, it doesn't offer anything special to facilitate action selection based on these Q-value estimates -- a non-trivial problem in itself when you use critic methods in continuous state spaces. In short, from the paper's current description, I couldn't deduce the advantages of the proposed method (or the problem characteristics where its advantages are especially noteworthy). - Positioning w.r.t. related work. Currently, the paper seems to mention only one three BRL works, and they are really dated by now. Exactly what other algorithms the proposed work should be compared to depends on its motivation, but one algorithm that is almost certainly relevant is BAMCP: Guez, Silver, Dayan. "Scalable and Efficient Bayes-Adaptive Reinforcement Learning Based on Monte-Carlo Tree Search". JAIR, 2013. Its vanilla version described above assumes discrete state and action spaces, but using discretization this restriction can be lifted, and its derivatives may have done so already. In general, there has been a lot of BRL work since references [7], [8], and [9], as described in the following survey: Ghavamzadeh, Mannor, Pineau, Tamar. "Bayesian Reinforcement Learning: A Survey". Foundations and Trends in Machine Learning, 2015. Newer versions are on arXiV. It would be good to see the related work coverage expanded to at least some of the recent BRL methods described there. More technical comments: - The phrase "uncertain dynamics" is somewhat unfortunate, because it can be easily misinterpreted to refer to dynamics that are probabilistic, while this paper seems to handle only the cases where the dynamics itself is near-deterministic but unknown. - The claim on lines 108-113 makes Q-value computation for a known \theta looks harder than it is. Every known \theta essentially encodes an MDP with known dynamics (and known reward function -- this appears to be assumed in the paper). Calculating the Q-function for this MDP is a planning problem, which can be solved in a wide variety of ways, which can be very efficient depending on the MDP structure. In particular, I'm not sure why deep learning is necessary here. Is this because you assume continuous states and actions and a DNN-based Q-value approximator? - Is Equation 3 missing the expectation sign? If not, it is missing something else, because, as stated, the value of the left-hand side depends on a particular trajectory sampled from \pi, and the LHS value is ill-defined without conditioning on this trajectory. - I'm confused by Section 3.1. Its stated goal is computing (an approximation of) Q^*_{\theta}, the Q-value function of the optimal policy under MDP parameters \theta. However, then it goes on to explain how to compute Q^{\pi}_{\theta} for policy \pi from Equation 10. What's the relationship between \pi from Equation 10 and the optimal policy under \theta? Line 147 talks about the exploration-exploitation tradeoff -- why do you need to resolve it if you are doing all of this for a _specific_ MDP parameter vector \theta? - The paper switches between subscript and superscript indexes for \theta. It appears to be using subscript indexes for \thetas sampled from the prior (as on line 175) and superscript indexes for \thetas sampled from the posterior (as in Equation 14). Is this the distinction the paper is trying to make? If so, I suggest keeping it uniform and using, e.g. \theta^{prior}_i and \theta^{posterior}_i instead. - Are there any hypotheses as to why the proposed algorithm outperforms MMRL on the synthetic MDP? - To beef up the the experimental evaluation, I'd replace the synthetic MDP with something more complicated. Language errors: "the sequence of observed joint state and action simulated" --> "the sequence of observed joint state and action pairs simulated" "specially in regions" --> "especially in regions" IN THE REBUTTAL, please comment on the motivation/advantages of the proposed algorithms w.r.t. others, particularly BAMCP and its derivatives, and cover the questions in the "technical comments" section of the review, as space allows.

Reviewer 2



The authors present a framework for MDP planning in a situation where 1) interaction with the system is limited, 2) the dynamics are uncertain but the reward function is known, and 3) the state space is continuous or smooth in some known way. The approach is to compute Q^* for a collection of transition dynamics θ drawn from a prior over transition dynamics. Each time the agent interacts with the system, a collection of θ are drawn from the posterior over transition dynamics. Rather than explicitly plan in each of these posterior transition models, the authors use a GP to estimate Q-values for the posterior θ from the Qs computed for the prior θ. These Qs can then be used to create a behaviour policy. The idea is essentially to build one uber-q-function parameterized by \theta that conditions on different transition models. This makes sense if the Q-functions vary mostly smoothly with \theta, which seems plausible. Quality: I believe that the technical aspects of the paper are well thought-through, and I think the experiments are adequate. Clarity: Overall I felt the paper was very well-presented. I have a few specific comments/questions about clarity. "The accuracy of the proposed framework is demonstrated using a small example as well as noisy synthetic gene expression data from a melanoma gene regulatory network model." - It might help to make it more clear what the control problems are here rather than just the subject area. "Let the uncertainty in the dynamics be encoded into a finite dimensional vector θ, where θ takes its value in a parameter space..." - I find this difficult to interpret, since later on θ seems to represent just a transition model. Could it be "...encoded into a random variable \Theta"? In the algorithm: "Draw N sample points" - Always try to be as specific as possible. "Draw N transition models"? Originality: The paper is a well thought-through combination of existing techniques, applied in a useful way. Significance: The work has clear interest within the RL community at NIPS, and will be of interest to people with applications that are more data-poor (as opposed to large-scale deep learning methods.) I think it makes a significant contribution to the ability to solve these kinds of problems. === Update I thank the authors for their response and clarification.

Reviewer 3



** summary ** This paper proposes to tackle the problem of solving Markov Decision Processes whose parameters are poorly known. The authors focus on situations where Reinforcement Learning cannot be used directly, neither through direct interaction with the environment nor via a simulator. For that they adopt a Bayesian framework. After having mentioned related works and the main advantages of the framework exposed in this paper, the definitions and notations specific to Markov Decision Processes are provided. The Bayesian policy optimization criterion for MDPs with imprecise parameters is then presented followed by the description of the proposed framework. An offline framework planner, using a Gaussian Process Temporal Difference (GPTM) learning algorithm, computes an optimal policy for several sampled parameters from the prior distribution. The online planner uses data collected online and an MCMC algorithm to sample posterior parameters. The values associated with the posterior parameters are also computed with the GPTM algorithm and allow the calculation of the optimal actions. This framework is tested against two other algorithms, namely Multiple Model-based RL and One-step lookahead policy, and with two different environments. The first environment contains a state variable and an action variable, both of which are continuous, and the imprecision concerns the transition function. The second environment is a belief-MDP, with three actions and the imprecision also concerns the transition function (via the belief-MDP observation function). The results show that the approach is better in terms of learning speed and policy optimality than the other two algorithms. ** quality ** This paper has a nice introduction, the authors look at an important problem, develop a framework adapted to the problem and test it on several systems. However, the paper lacks theoretical results or risk analysis of the produced policies. ** clarity ** This work can be read and understood quite easily because it is mostly clear. However most of the equations on pages 4 and 5 are not very readable and understandable. They should be simplified and accompanied by intuitive and educational remarks and explanations, or even a visual illustration. Other less important remarks that would improve clarity follow. page 4: Equation 7 is sticking out. page 5: The variances are sticking out. page 7: The notations "N=10" and "N^{post}=100" could be reused < As expected, the minimum average reward > As expected, the maximum average reward OBDM is not defined (what's the O for?) page 8: < in a simplex of size 128 > in a simplex of size 127 ** originality ** The authors have done a good bibliographical work to position their study in the relevant literature. However, work has also been carried out with different models of uncertainty. There are other models taking into account the mentioned imprecision which is unfortunately often present in practice. For example work was produced for planning under Knightian Uncertainty, (Trevizan "Planning under Risk and Knightian Uncertainty." IJCAI 2007), with Dempster-Shafer belief Theory (Merigo "Linguistic aggregation operators for linguistic decision making based on the Dempster-Shafer theory of evidence." International Journal of Uncertainty, Fuzziness and Knowledge-Based Systems 2010), with Possibility Theory (Drougard "Structured Possibilistic Planning Using Decision Diagrams" AAAI 2014), and with Imprecise Probability Theory (Itoh "Partially observable Markov decision processes with imprecise parameters." Artificial Intelligence 2007). Some of these references should be cited since they address the same problem. ** significance ** Experimental results are quite impressive and the motivation of this framework, i.e. the fact that reinforcement learning is not always applicable and that the parameters of a MDP are often imprecise, is of great importance in practice. However, the use of a Bayesian framework requires the knowledge or the choice of a prior distribution. The framework of this paper also requires the choice of N and Npost. How are these choices made in practice?